# Batch effect-invariant graph neural networks for predicting chemotherapy response in triple-negative breast cancer patients

**Asif Khan** [* 1]   **Luciana Luque** [* 2]   **Giussepe Torisi** [* 2]   **Claudia Owczarek** [3]   **Maddy Parsons** [3]   **Chris Sander** [1 4]
**Linus Schumacher** [2 4]

## Abstract

Triple-negative breast cancer (TNBC) is a particularly aggressive subtype of breast cancer that is usually treated with chemotherapy. However, the effectiveness of the treatment can vary widely. Accurate prediction of the response to chemotherapy is crucial in preparing effective personalized treatment. This paper introduces a machine learning framework that uses imaging mass cytometry (IMC) data from clinical trials to train graph neural networks (GNNs) to predict whether a patient will respond to chemotherapy. Our approach combines single-cell protein expression and spatial cell-cell contact information extracted from IMC images. To account for staining variability known as batch effects, we introduce a surrogate loss function that enables learning of a representation space predictive of response, yet invariant to batch artefacts. We investigate different graph construction methods (k-nearest neighbors, k-atmost neighbors, Delaunay triangulation) to capture cell-cell contact delineating tumor microenvironment. Our framework demonstrates improved predictive performance through batch effect correction and effective integration of protein expression with spatial cellular relationships.

## 1. Introduction

Breast cancer is a prevalent global form of cancer, posing significant challenges in treatment and management due to high incidence and mortality rates worldwide (Sung et al., 2021). Among its subtypes, TNBC is particularly aggressive, characterized by the lack of estrogen receptor (ER), progesterone receptor (PR), and human epidermal growth factor receptor 2 (HER2) (Bianchini et al., 2016). ER, PR and HER2 are molecular targets for specific targeted therapies in breast cancer. Their absence in TNBC means that these targeted therapies, such as anti-estrogen (e.g., tamoxifen) or anti-HER2 (e.g., trastuzumab) drugs, are ineffective for treating TNBC patients (Shetti et al., 2019). With limited targeted therapeutic options TNBC treatment often relies on conventional chemotherapy. However, there is significant variability in the response rates to chemotherapy, resulting in unnecessary toxicity for non-responders (Bianchini et al., 2016). Predicting a patient's response to chemotherapy is crucial for personalized treatment planning, enabling the identification of alternative treatments when necessary.

Recent advancements in spatially resolved imaging technologies, such as IMC, have made it possible to quantify multiple protein markers within individual cells, a promising development for designing cancer therapeutics. IMC uses metal-tagged antibodies to detect and quantify over 40 proteins or other molecules in biological samples, with a spatial resolution of 1 $\mu m$ and an ablation frequency of 200 Hz (Giesen et al., 2014). This technology offers a powerful tool for fast profiling of selected areas of biopsy samples, enabling studies ranging from spatial analysis of tumour microenvironments to the characterization of pathological features in diseases such as TNBC.

GNNs have demonstrated promising capabilities in capturing spatial relationships and modeling complex biological networks (Xu et al., 2019; Li et al., 2022). GNNs are inherently designed to handle graph-structured data, making them a suitable candidate to integrate protein expression and cell-cell contact information to learn features that capture disease development (Wang et al., 2021).

In this paper, we utilized an IMC dataset of patients to develop a batch effect-invariant GNN model for predicting response to chemotherapy. The IMC images were processed through a dedicated preprocessing pipeline to extract single-cell protein expression levels and their spatial information. However, a significant challenge in analyzing these spatial single-cell protein data is the high variability resulting from

[*]Equal contribution [1]Department of Systems Biology, Harvard Medical School, Boston, USA [2]Centre for Regenerative Medicine, University of Edinburgh, Edinburgh, UK [3]Randall Centre for Cell and Molecular Biophysics, King's College London, Guy's Campus, UK [4]Joint Supervision. Correspondence to: Asif Khan <asif.khan@hms.harvard.edu>.

*Accepted at the 1st Machine Learning for Life and Material Sciences Workshop at ICML 2024.* Copyright 2024 by the author(s).

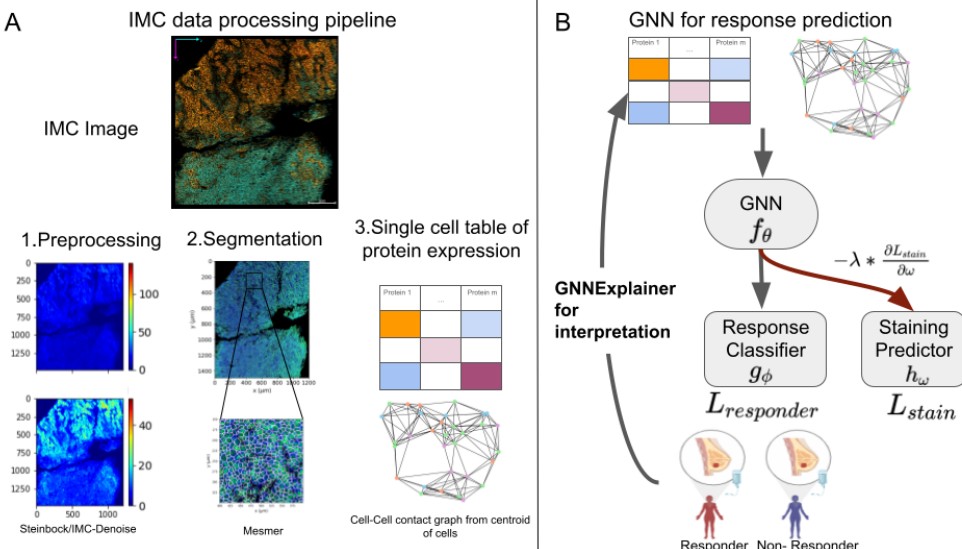

Figure 1. We present a pipeline that includes feature extraction from raw IMC images and building a response prediction model. On the left is a feature extraction stage, where the first step involves using Steinbock (Windhager et al., 2023) and IMC-Denoise (Lu et al., 2023) methodologies to mitigate artefacts such as hot pixels and shot noise. Subsequently, cell segmentation is performed using Mesmer (Greenwald et al., 2022), where the centroids of identified cells denote their spatial coordinates. These centroids are then utilised to construct a graph, wherein the mean pixel intensities per cell are used to quantify its protein expression. This protein expression data is integrated for training GNN as showcased on the right. Following model training, GNNExplainer is deployed to find the most predictive protein profiles indicative of a positive case on a held-out patient data.

the staining with antibodies, making it difficult to separate relevant biological signals, such as patient-to-patient differences, from noise. To mitigate such batch effects, we introduced a surrogate loss that helps in learning a representation space that is predictive of response while being invariant to batch-specific artefacts.

To summarise the key contributions of our work are:

1. **Response prediction with batch correction.** We propose a learning objective that uses protein expression and cell-cell contact graphs to learn a representation that is predictive of target class while erasing information that can be predictive of batch artefacts. We achieve this by introducing a surrogate loss for "batch effect" prediction and when updating a GNN we reverse the gradients from this predictor. Our results show improved predictive performance with such a regularization objective.

2. **Effect of different graph construction methods on response prediction.** Given spatial coordinates of cells there are different ways to construct a graph representation to capture cell-cell contact relationships informative of tumor structure and tumor-microenvironment composition. We investigate three choices: k-nearest neighbor (k-NN), k-atmost neighbors, and delaunay triangulation. We show for IMC data k-NN achieves best performance compared to other graph choices.

## 2. Methodology

We start by preprocessing IMC images to extract protein expression profiles and centroids of individual cells. Then, we introduce our method that uses a GNN to predict whether patients will respond to chemotherapy, while also removing batch effects using a surrogate optimization objective. The complete framework for predicting the response of TNBC patients to chemotherapy based on IMC imaging data is illustrated in Figure 1. Next, we introduce the real-world dataset of IMC images utilized in our experiments.

**TNBC IMC Dataset** We use a real-world dataset generated as part of the Wellcome Leap Delta Tissue program [1], mostly from retrospective samples from a large number of patients (via biobanking consent) and a few from the FORCE clinical trial. A cohort of patients diagnosed with triple negative breast cancer (TNBC) was recruited and tissue biopsy taken before starting neoadjuvant chemotherapy. A pathologist identified a set of regions of interest (ROIs) based on H&E staining, which are then analyzed with IMC. Here, IMC measures protein abundance using a panel of 35 metal-tagged antibody markers for tumoral, immune, and stromal cells (CD3, CD4, CD8a, CD11b, CD14, CD16, CD20, CD27, CD31, CD38, CD44, CD45, CD45RO, CD68, CD107a, CD163, CD366, Beta-Catenin, E-Cadherin, Pan-

---

[1] https://wellcomeleap.org/delta-tissue/

Keratin, Vimentin, Tbet, FOXP3, HLA-DR-DQ-DP, Alpha-SMA, Granzyme-B, B7-H4, Ki-67, PD1, PD-L1, PD-L2, p53, Collagen Type I, EGFR, VEGF) as well as two antibodies for DNA[2]. This results in an image with 35 channels (one for each protein marker).

Patients are classified based on their residual cancer burden (RCB) score at the end of the treatment. Here, we adopt a binary label for the patient response, defining a pathological, complete responder (pCR) if the RCB is 0 and non-responder otherwise (nR). We implement strict quality control metrics on cells and remove ROIs with less than 1000 cells. From 445 ROIs and 58 Patients, the dataset that passes quality control consists of 396 ROIs from 51 patients associated with five staining groups. In the dataset there are 37 nR and 24 pCR patients.

**Data preprocessing**   The IMC data contain hot pixel noise and shot noise primarily resulting due to the detection mechanism. Hot pixel noise occurs due to the formation of antibody aggregates that produce regions of the image with high antibody counts. Shot noise arises because of the discrete nature of ion detection and antibody binding, which causes random fluctuations in signal intensity. These noise sources collectively make it challenging to determine protein expression levels with high precision. As part of the pre-processing step, we utilize Steinbock (Windhager et al., 2023) and a state-of-the-art deep-learning-based algorithm IMC-Denoise (Lu et al., 2023) to extract and denoise the images. Subsequently, we apply Mesmer (Greenwald et al., 2022), a segmentation algorithm to delineate individual cells and quantify protein expression levels within distinct cellular regions. Mesmer is a deep learning-based algorithm that provides cell masks for spatial localization of proteins. Once the cells are segmented, centroids are computed to represent the cells' spatial coordinates within an ROI.

**Graph construction**   The cell-cell contact graph captures cellular interactions, providing insights into the spatial connectivity of cells at varying scales. In our work, we combine this graph-based representation with protein expression data to predict response to chemotherapy. Here, we provide a detailed description of different methods for the graph construction.

We use three distinct graph construction approaches: k-nearest neighbours (kNN), k-almost neighbours, and Delaunay triangulation. The kNN method connects each cell to its k nearest neighbours based on the Euclidean distance between centroids. In contrast, the k-atmost neighbour approach uses a distance threshold. It connects each cell with up to k cells if their distance falls within the specified thresh-

---

[2]DNA is only used for segmentation. Since it is expressed in all cells we exclude it from prediction.

old while ensuring each cell is at least connected to one nearest neighbour. The threshold here is the weight that separates the weakest $(1 - \frac{k}{N}) \times 100\%$ of edges (where N is the number of nodes in a graph). Finally, the Delaunay triangulation method connects the centroids of cells forming triangles such that no cell centroid lies inside the circumcircle of any triangle, inferring cellular adjacency based on their spatial arrangement.By incorporating these graph construction techniques, we aimed to capture the potential cellular interactions and investigate their performance on the downstream task of response prediction.

**Prediction of response to chemotherapy**   The experimental protocol for data generation introduces technical artefacts that may obscure biological features. For example, differences in sample preparation or batches of reagents may introduce variations in the measured protein abundance that is unrelated to the biological variability of the samples. These artefacts affect the measurements in groups or batches and are known in the literature as batch effects. These batch effects ultimately bias the distribution of measurements. Correcting batch-related variations is crucial to disentangle biological signals from technical artefacts. To this end, we introduce an optimization objective using a gradient reversal to simultaneously predict responder labels while censoring information that includes batch-sensitive information.

Let $\mathbf{X} \in \mathbb{R}^{n \times d}$ be a protein expression matrix, $G = (V, E)$ be cell-cell contact graph with $n = |V|$ as a set of vertices and $|E|$ as a set of edges (constructed as described in Section 2), $d$ is number of protein markers, $s \in S$ be a staining label where unique labels are $S = \{0, 1, 2, 3, 4\}$, and $y \in Y$ be a label that takes value 1 for responder or 0 for non-responder. We use a GNN encoder $f_\theta$ to embed input $G$ and $\mathbf{X}$ of each ROI to a fixed length representation $\mathbf{z} \in \mathbb{R}^{d_z}$ where $d_z$ is a dimensionality of latent space, $g_\phi$ to predict responder/non-responder labels, and $h_\omega$ to predict the staining labels. To encourage the encoder to learn features that are not sensitive to staining labels, the encoder parameters are optimized to maximize the loss $L_{\text{stain}}(.,.)$ and simultaneously minimize the loss of response predictor $L_{\text{responder}}(.,.)$. In Equation,

$$
\begin{aligned}
\mathcal{L} = \min_{\theta, \omega, \phi} \; &L_{\text{responder}}(g_\phi(f_\theta(G, \mathbf{X})), y) \\
&- \lambda L_{\text{stain}}(h_\omega(f_\theta(G, \mathbf{X})), s)
\end{aligned}
$$

where $\theta$, $\phi$ and $\omega$ are the parameters of the encoder, response predictor, and batch predictor, respectively. The final loss is averaged over all ROIs in the dataset. The optimization is performed using a gradient reversal layer (Raff & Sylvester, 2018) that in backward pass negates the gradient w.r.t the staining objective and in forward pass acts as an identity function. This optimization mechanism allows the

GNN encoder to learn representations that are invariant to the staining effect and contain the necessary information required for the response prediction task. Response prediction is a binary task for which $L_{\text{responder}}$ reduces to a binary cross-entropy loss, and stain prediction is a multiclass classification that reduces $L_{\text{stain}}$ to a categorical cross-entropy loss.

**ROI to patient-level prediction.** Pathologists annotate ROIs, which can be stained in different batches for the same patient. We, therefore, treat them as separate samples for training purposes. However, predicting a patient-level response is more relevant for any clinical decision-making. During the evaluation phase, patient-level predictions were obtained using a majority voting scheme across the ROIs belonging to the same patient. Specifically, let $R_p = \{r_1, r_2, \ldots, r_n\}$ represent the set of ROI-level predictions for patient $p$, where $r_i \in \{0, 1\}$ represents the predicted response (0 for non-responder, 1 for responder) for the $i^{th}$ ROI. The patient-level prediction $P_p$ was then determined as:

$$P_p = \begin{cases} 1, & \text{if } \sum_{i=1}^{n} r_i > n/2 \\ 0, & \text{otherwise} \end{cases} \quad (1)$$

### 2.1. Experimental setup

We implemented our framework in Python using PyTorch (Paszke et al., 2017) and PyTorch Geometric library (Fey & Lenssen, 2019) for GNNs. The encoder architecture comprises a two-layer Graph Convolutional Network (GCN) (Kipf & Welling, 2017) followed by average pooling. The response prediction head is a linear layer that predicts the probability of a patient's response to chemotherapy. In contrast, stain prediction is a classification layer that predicts the probability of an ROI coming from a specific staining label. The regularization hyperparameter $\lambda$ was set to $0.25$ to balance the loss contribution.

To evaluate the predictive performance of the model, we report the area under the curve (AUC), accuracy, and F1 score to capture the effect of class imbalance. We report these scores for evaluation of ROI-level prediction and patient-level prediction using majority voting. These metrics comprehensively evaluate the model's ability to predict patient response to chemotherapy in TNBC.

We split our dataset into a training and test set with a 70% and 30% split. We set k=7 for the k-NN and k-atmost graph construction. We experimented with different k values from the set $\{1, 2, 3, 4, 5, 6, 7, 8, 9, 10\}$ and found that k=7 resulted in the best performance.

## 3. Results and Discussion

**Ablation comparing different graph construction methods.** When working with the spatial coordinates of cells within an ROI, there are different methods available for creating a graph representation. Table 1 summarizes the performance of different graph construction approaches. Our results indicate that the k-NN graph outperforms other approaches in terms of ROI-level performance and achieves comparable performance to the Delaunay triangulation method at the patient level. Owing to its simplicity and superior performance, we use the k-NN approach for graph construction in the subsequent analyses presented in this paper.

*Table 3.* Ablation of batch effect correction with k-NN graph (k=7). Introducing staining loss leads to improved performance, encouraging the GNN to learn representations invariant of batch effects.

| | Batch Correction | | No Batch Correction | |
| --- | --- | --- | --- | --- |
| | ROI | Patient | ROI | Patient |
| AUC | **0.797** | **0.928** | 0.766 | 0.857 |
| Accuracy | **0.751** | **0.947** | 0.766 | 0.894 |
| F1 Score | **0.752** | **0.923** | 0.739 | 0.833 |

**Ablation of staining correction.** To investigate the benefit of including a staining predictor, we run an ablation study where we compare the performance of the model trained with and without the staining predictor. The performance is reported in Table 2. We observe training with our objective significantly improves the performance with an improvement of $3.1\%$ in ROI-level and $7.1\%$ in patient-level AUC, demonstrating the benefit of the proposed approach.

**Comparison to baseline.** To demonstrate the benefit of incorporating spatial information, we compare our model with baselines such as logistic regression, random forest, and XGBoost. The results of our model are outlined in Table 2 at ROI and patient level prediction. GNN model consistently outperforms the baselines demonstrating the benefit of spatial connectivity information from cell-cell contact graph.

**Feature attribution.** We utilized the GNNExplainer (Ying et al., 2019) algorithm to assess the importance of protein markers in the prediction of responders. GNNExplainer explains GNN predictions by identifying the most relevant node features and sub-graph structures. It determines attribution by computing the mutual information between the input (a subset of node attributes and sub graphs) and the GNN's prediction, effectively quantifying the contribution of each node feature to the final prediction. Graph level contribution is obtained by summing up scores of each marker across all nodes. Figure 2 reports the top 10 protein markers

*Table 1.* Comparison of patient-level predictions using different graph construction approaches. k-NN achieves the best ROI-level prediction performance and the same performance as Delaunay on patient-level predictions.

| Graph Type | AUC | | Accuracy | | F1 Score | |
|---|---|---|---|---|---|---|
| | ROI | Patient | ROI | Patient | ROI | Patient |
| K-NN (K=7) | **0.797** | 0.928 | **0.751** | 0.947 | **0.752** | 0.923 |
| K-atmost neighbors (K=7) | 0.698 | 0.785 | 0.657 | 0.842 | 0.649 | 0.723 |
| Delaunay triangulation | 0.774 | 0.928 | 0.704 | 0.947 | 0.698 | 0.923 |

*Table 2.* Here, we demonstrate the benefit of integrating cell-cell contact graph information with protein expression levels for predicting response to chemotherapy. Results show improved performance compared to baseline methods trained solely on protein expression features.

| Method | Logistic Regression | | XGBoost | | Random Forest | | GCN | |
|---|---|---|---|---|---|---|---|---|
| | ROI | Patient | ROI | Patient | ROI | Patient | ROI | Patient |
| AUC | 0.610 | 0.571 | 0.531 | 0.571 | 0.541 | 0.571 | **0.797** | **0.928** |
| Accuracy | 0.583 | 0.684 | 0.590 | 0.684 | 0.597 | 0.684 | **0.751** | **0.947** |
| F1 Score | 0.534 | 0.571 | 0.485 | 0.25 | 0.544 | 0.250 | **0.752** | **0.923** |

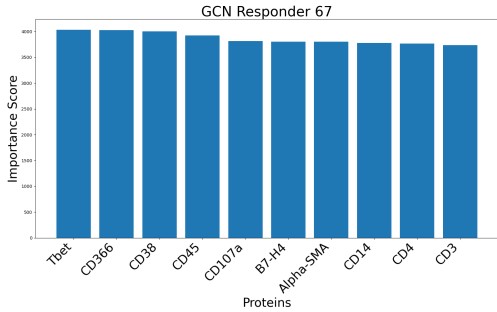

*Figure 2.* We rank all protein markers by their importance score determined by GNNExplainer algorithm and report top 10 markers and their score for response prediction.

for a patient who responds to therapy.

Most of the markers in the top ten are immune markers, with the exception of B7-H4 and Alpha-SMA. There has been recent evidence suggesting that TNBC tumours with higher tumour infiltrating lymphocytes (TILs) are associated with a better prognosis and a higher likelihood of achieving pathological complete response (Huertas-Caro et al., 2023). In particular, Tbet, which regulates effector T-cell activation, has been identified as a better prognostic indicator for TNBC (Mori et al., 2019). Additionally, while there is some evidence that the overexpression of B7-H4 leads to a poor prognosis in TNBC (Wang et al., 2018), a recent study indicates that the loss of B7-H4 expression in breast cancer cells escaping from T cell cytotoxicity contributes to epithelial-to-mesenchymal transition (Zhou et al., 2023), which is known to have a poor prognosis.

**Discussion** Developing a model that can predict the re-

sponse of patients to chemotherapy has implications for personalized treatment strategies. By identifying patients who are not responding, a more personalized therapeutic treatment can be designed. Patients who are predicted to respond favourably to a specific treatment can be prioritized for that therapy, maximizing the likelihood of positive outcomes and minimizing unnecessary exposure to ineffective, costly treatments as well as toxicity and side effects. Conversely, patients predicted to be non-responders can be promptly transitioned to alternative therapeutic options, avoiding delays in effective treatment and potentially improving overall survival rates (Glasson et al., 2023). In future work, we aim to combine our predictions with clinicians to take a step towards the effective translation of IMC-derived insights into clinical practice.

## 4. Conclusion

This paper introduced a GNN-based approach for predicting chemotherapy response in TNBC patients using a novel IMC dataset. A key challenge addressed was mitigating batch effects from staining variability that can obscure biologically relevant signals. We proposed a surrogate objective function implemented via gradient reversal, allowing the model to learn a representation space predictive of response while invariant to batch artefacts. Our framework demonstrates the potential of GNNs in using spatially-resolved IMC data for accurate response prediction by integrating spatial context and protein expression features within a unified graph-based framework. This approach offers a promising avenue for advancing personalized treatment strategies and improving clinical outcomes for TNBC patients through effectively integrating multichannel imaging data.

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
