# OpenReview forum: "Batch-effect invariant graph neural networks for predicting chemotherapy response in triple-negative breast cancer patients"
_ICML.cc/2024/Workshop/ML4LMS — ML4LMS Poster_

### Official Review · Reviewer_Lhhr · 2024-06-08
**The proposed method and results seem promising, further comparisons are still needed**

**Rating:** 6
**Confidence:** 4

**Review:**

This paper proposes a novel approach to predict treatment response in Triple-negative breast cancer (TNBC) patients based on imaging mass cytometry (IMC) data. The authors leverage different existing tools to extract single-cell protein expression values as well as cell spatial information and use such data to train a graph neural network (GNN) for classification with a modified loss function to also account for the presence of batch artifacts.

**Pros**
- The paper is well-written and structured. The proposed framework and performed experiments are clearly described.
- The application case considered is of interest.

**Cons**
- Only the GNN-based model is evaluated in this work. Comparison with other simpler models/architectures could additionally demonstrate the effectiveness of the proposed approach.

**Minors**
- Possibly due to the limited number of samples available, the authors use the final test set to select the best graph construction method. I would still recommend using cross-validation (CV) to select the graph type. Moreover, a nested CV approach could be used to overcome the limitations related to the small sample size.

---

### Official Review · Reviewer_2cgC · 2024-06-12

**Rating:** 5
**Confidence:** 3

**Review:**

Pros:

1. The paper is well-written, and the motivation behind using GNN to predict chemotherapy response from imaging mass cytometry data is sound and well-justified.

2. The paper acknowledges and addresses the issue of batch effect in the data, which is a critical factor in biological data.

Cons:

1. The idea of removing batch effects in biological data and the use of adversarial loss or gradient reversal for this purpose is not novel.

2. The paper does not include comparisons with other baseline methods, which makes it difficult to gauge the effectiveness of the proposed approach relative to existing solutions.

3. Minor point: the resolution of Figure 1 is low.